# Enhancing care quality and accessibility through digital technology-supported decentralisation of hypertension and diabetes management: a proof-of-concept study in rural Bangladesh

Wubin Xie,[1] Rina Rani Paul,[2] Ian Y Goon,[3,4,5] Aysha Anan,[2] Aminur Rahim,[5] Md Mokbul Hossain ![ORCID],[2] Fred Hersch,[6] Brian Oldenburg,[4,7] John Chambers,[1,8] Malay Kanti Mridha ![ORCID] [2]

For numbered affiliations see end of article.

**Correspondence to**
Professor Malay Kanti Mridha;
malay.mridha@bracu.ac.bd

## ABSTRACT

**Objective** The critical shortage of healthcare workers, particularly in rural areas, is a major barrier to quality care for non-communicable diseases (NCD) in low-income and middle-income countries. In this proof-of-concept study, we aimed to test a decentralised model for integrated diabetes and hypertension management in rural Bangladesh to improve accessibility and quality of care.

**Design and setting** The study is a single-cohort proof-of-concept study. The key interventions comprised shifting screening, routine monitoring and dispensing of medication refills from a doctor-managed subdistrict NCD clinic to non-physician health worker-managed village-level community clinics; a digital care coordination platform was developed for electronic health records, point-of-care support, referral and routine patient follow-up. The study was conducted in the Parbatipur subdistrict, Rangpur Division, Bangladesh.

**Participants** A total of 624 participants were enrolled in the study (mean (SD) age, 59.5 (12.0); 65.1% female).

**Outcomes** Changes in blood pressure and blood glucose control, patient retention and patient-visit volume at the NCD clinic and community clinics.

**Results** The proportion of patients with uncontrolled blood pressure reduced from 60% at baseline to 26% at the third month of follow-up, a 56% (incidence rate ratio 0.44; 95% CI 0.33 to 0.57) reduction after adjustment for covariates. The proportion of patients with uncontrolled blood glucose decreased from 74% to 43% at the third month of follow-up. Attrition rates immediately after baseline and during the entire study period were 29.1% and 36.2%, respectively.

**Conclusion** The proof-of-concept study highlights the potential for involving lower-level primary care facilities and non-physician health workers to rapidly expand much-needed services to patients with hypertension and diabetes in Bangladesh and in similar global settings. Further investigations are needed to evaluate the effectiveness of decentralised hypertension and diabetes care.

## STRENGTHS AND LIMITATIONS OF THE STUDY

⇒ The study is among the first to evaluate a digital technology-supported, decentralised primary care model for integrated diabetes and hypertension management in a resource-constrained setting.

⇒ Generated important preliminary effectiveness data on a decentralised model to rapidly expand primary care services to people with hypertension and diabetes.

⇒ Enabled by digital tools, rich longitudinal data were collected to evaluate patient treatment outcomes and retention.

⇒ The duration of the study is short, and generalisability is limited by a local sample drawn from a subdistrict in northern Bangladesh.

death and disability. An estimated 10.5% (or 536.6 million) of adults are living with diabetes; CVD caused 18.6 million deaths worldwide in 2019.[1] Close to 80% of CVD deaths occurred in low-income and middle-income countries (LMICs).[2 3] The number of people with hypertension, a highly prevalent major risk factor for CVD, doubled from 1990 to 2019.[4] Complications of persistently elevated blood pressure and poor glycaemic control, such as heart disease, stroke, chronic kidney disease and blindness, pose enormous socioeconomic burdens on individuals, families and healthcare systems, contributing to poverty, inequality and social instability.[5] Despite the availability of effective and inexpensive medications and other management options for diabetes and hypertension for many years, treatment and control rates of these conditions remain unacceptably low in LMICs, particularly in South Asia.[6 7] Recent estimates indicate that in South Asia, only 26% of men and

## INTRODUCTION

Cardiovascular disease (CVD) and type 2 diabetes are major causes of premature

37% of women with hypertension were on treatment, and control rates were only 11% and 17% for men and women, respectively.[4]

Traditionally designed to provide basic episodic care with an emphasis on maternal and child health and infectious diseases, primary healthcare systems in LMICs are under-resourced and ill-prepared for the surging burden of CVD and other non-communicable diseases (NCD).[8] The critical shortage of healthcare workers, particularly in rural areas, is a major barrier to quality care.[9] Most evidently in grassroot-level facilities, primary care providers are untrained and unequipped to address primary and secondary prevention of NCD.[10 11] As a result, NCD care is poorly accessible in primary care systems.[12] Long waiting times and high costs associated with travel and lost wages discourage people from seeking care from qualified providers.[13–16]

Although few examples exist for hypertension and diabetes care, decentralisation with task shifting (sharing) in HIV care has been identified as a viable strategy to rapidly scale up antiretroviral therapy in resource-limited settings.[17–19] In this model of care, stable patients are transferred through a "down-referral" system from the doctor-managed clinic where they initiated care to non-physician health worker-managed local primary health clinics for continued monitoring and treatment. Decentralised care improves geographical accessibility, reduces costs associated with care seeking and enables the health system to serve a much larger patient population.[17] Local health service providers could monitor patients more effectively and efficiently and form a personal relationship that strengthens effective communication and counselling for treatment management.[20] People with HIV/AIDS managed locally are less likely to be lost to follow-up and more satisfied with care compared with those managed in centralised clinics.[21]

Digital technologies offer the potential to improve access to and quality of healthcare in LMICs.[22] Digital tools have been tested to support quality improvement in hypertension and diabetes management in South Asian settings. Several pilot studies have documented the utility of digital tools to facilitate protocol-based algorithmic care through clinical decision support and to promote continuity and coordination of patient visits across different healthcare settings, although findings are somewhat mixed.[23–25]

Little is known whether decentralising routine hypertension and diabetes services to lower-level primary care facilities and non-physician health workers is beneficial for enhancing care quality and accessibility in resource-poor settings and how digital technology can be harnessed to support the processes. In the present proof-of-concept study, we aimed to (1) generate preliminary data on the effectiveness of a novel digital technology-supported decentralisation of primary care diabetes and hypertension management in a rural subdistrict in Northern Bangladesh and (2) understand its potential future implications for the patient care pathway, the patient load in primary care facilities at a subdistrict and village level and patient retention.

## METHODS

### Context

Like many other LMICs, Bangladesh is experiencing rapid demographical and epidemiological transitions. In recent years, CVDs have replaced infectious diseases and maternal and neonatal conditions as the leading causes of death in Bangladesh.[26] The prevalence of hypertension and diabetes is high and has continued to increase in recent years.[27] The 2017–2018 Bangladesh Demographic and Health Survey showed a 9.8% prevalence of diabetes and a 27.4% prevalence of hypertension among the adult population.[28 29] Treatment and control rates remained low. Only 31.1% of adults with hypertension and 28.2% of adults with diabetes were on treatment, merely 12% achieved blood pressure control and 7% had controlled diabetes.[29 30]

Bangladesh has a three-tiered (subdistrict, union and ward) primary healthcare system in rural areas (figure 1). The subdistrict-level Upazila Health Complexes, typically 50-bed hospitals, are at the top of the primary care system, serving a population of 3000–4000. Community clinics are the most peripheral healthcare facilities and the first point of contact of the public healthcare system for rural residents. With two rooms and drinking water and lavatory facilities under a covered waiting area, community clinics were designed to be accessible to more than 80% of the rural population within less than 30-min walking distance. Staffed by a community healthcare provider with support from a few community health workers, community clinics have played critical roles in reducing malnutrition and child mortality and improving maternal and child health.[31]

Since 2011, the Government of Bangladesh has promoted improved hypertension and diabetes care through national multisectoral actions, most notably through the establishment of dedicated NCD-related healthcare delivery points ('NCD Corners') at subdistrict hospitals.[32] Although originally planned to involve lower-level primary care facilities in the delivery of NCD care, hypertension and diabetes care are largely unavailable at the village level.[33]

### Study settings

This proof-of-concept study was implemented in a newly initiated study NCD clinic at the Parbatipur subdistrict, in collaboration with three community clinics at Khorakhai, Jahanabad and Manmathapur. Parbatipur is a subdistrict of Dinajpur district, in the Rangpur Division of northern Bangladesh. Parbatipur had a population of 365 103 in 2011. The public primary healthcare system in Parbatipur consists of one Upazila Health Complex (subdistrict hospital), 10 union health (sub)centres and 34 community clinics.[34] With a maximum capacity of seeing 40–50 patients per day, the NCD Corner at the Parbatipur

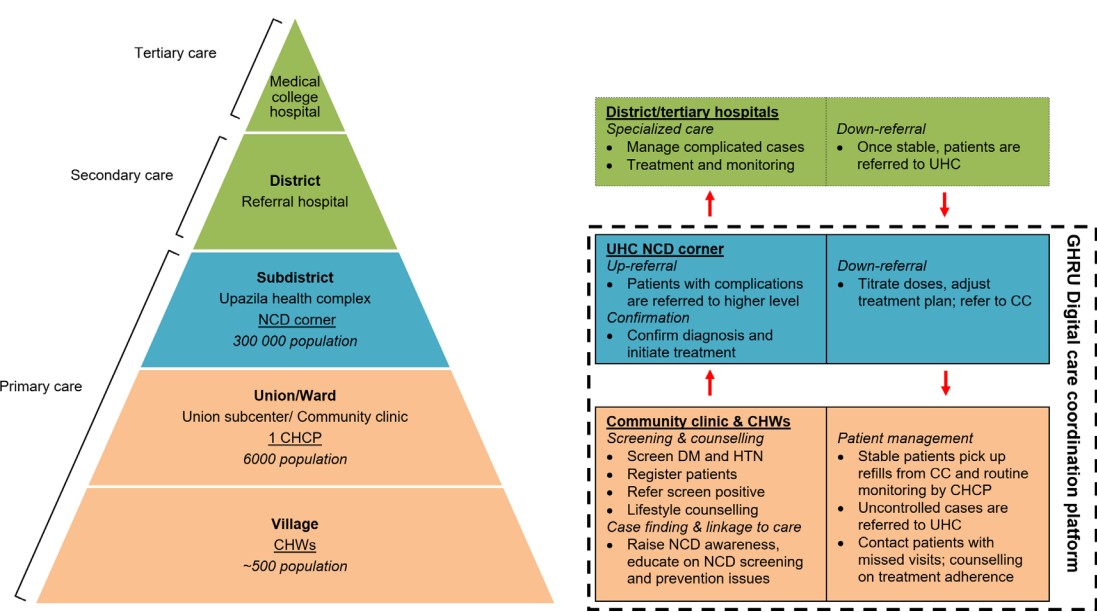

**Figure 1** NCD service delivery hierarchy and coordinated NCD care. CC, community clinic; CHCP, community healthcare provider; CHW, community health worker; GHRU, global health research unit; NCD, non-communicable disease; UHC, Upazila Health Complex. *Government community health workers include family welfare assistants, health assistants and multipurpose health volunteers; however, a designated community health worker was recruited and trained for NCD-related tasks in the present study. Union subcentres were not involved in this present study. Union subcentres are at a higher level of hierarchy than community clinics.

Upazila Health Complex manages a few hundred diabetes and hypertension patients—a very small fraction of the estimated patient population with these conditions.

### Study design

The study is a single-cohort proof-of-concept study. Decentralised hypertension and diabetes care comprised screening, routine monitoring and dispensing of medication refills at village-level community clinics, as well as treatment initiation and titration at subdistrict-level NCD clinics. Patient flow in this model of care included community health workers motivating adults aged 40 years and above to get screenings for hypertension and diabetes at community clinics; once confirmed, patients received initial care at the study NCD clinic and subsequent follow-up at community clinics. At the follow-up visits, community healthcare providers perform a basic physical examination (ie, body weight and height measurement, calculation of body mass index (BMI), blood pressure and blood glucose measurement), inquire about any new symptoms, assess CVD risk, emphasise the importance of medication adherence, provide advice regarding a healthy lifestyle and refill prescribed medications. Patients were referred back to the NCD clinic for medication titration and further evaluation if the treatment target was not met or any urgent symptoms or conditions were recorded.

The study NCD clinic was staffed by one medical doctor and one paramedic. Both the doctor and the paramedic received a 4-day training on hypertension and diabetes management following the package on essential NCD interventions of the WHO. The paramedic had 1 year of

postgraduate medical education. The doctor carried out a physiological examination and provided diagnostic and treatment services according to the Bangladesh National Protocol for Management of Diabetes and Hypertension, adopted from the WHO Package of essential NCD interventions.[35] The paramedic was responsible for taking medical histories, measuring blood pressure and blood glucose, delivering health education, counselling and medication disbursement.

One community healthcare provider was employed to provide services in three community clinics (2 days per week in each community clinic). The community healthcare provider was supported by three newly recruited NCD community health workers (one for each community clinic). The community healthcare provider received a 1-year postsecondary education medical training. Community health workers were newly recruited from local communities with a minimum of secondary school education. The community healthcare provider and the community health workers received a 7-day initial training on behavioural risk factors of hypertension and diabetes, anthropometry, lifestyle counselling and follow-up, and received ongoing regular (weekly to monthly) supervision from the study NCD clinic medical staff and the local study investigators.

The community healthcare provider and community health workers were responsible for administering written informed consent, registering participants and referring adults for screening, follow-up, medication adherence and health education/counselling in the community. They were regularly supervised by the medical doctor

from the study NCD clinic and by the study investigators. The study NCD clinic and community clinics were equipped with the necessary medical, pharmaceutical and logistical supplies and equipment. Electronic health records were maintained by the study digital care coordination platform.

## Patient screening and eligibility criteria

Adults aged 40 and above were approached at their households and registered and referred by community health workers to be screened at the community clinics. Those with a history of hypertension or diabetes and who screened positive for hypertension or diabetes were referred to the study NCD clinic for further evaluation. The WHO non-laboratory-based CVD risk scores were calculated.[35] Existing and new patients with hypertension and/or diabetes were enrolled. Diabetes was diagnosed if any of the following criteria were met: (1) the participant reported taking medications for diabetes; (2) fasting plasma glucose (FPG) ≥7.0 mmol/L or random plasma glucose (RPG) ≥11.1 mmol/L with a symptom of diabetes at one visit and (3) FPG ≥7.0 mmol/L or RPG ≥11.1 mmol/L at two separate visits, at least 1 day apart. Hypertension was diagnosed if any one of the following criteria was met: (1) participant reported taking medications for hypertension; (2) average systolic blood pressure (SBP) ≥160 mm Hg and/or diastolic blood pressure (DBP) ≥100 mm Hg on a single visit; (3) average SBP ≥130 mm Hg and/or DBP ≥80 mm Hg on a single visit and participant had a high risk of CVD (10-year CVD risk score ≥0.3) or participant reported a diagnosis of diabetes and (4) average SBP ≥140 mm Hg and/or DBP ≥90 mm Hg on two separate visit, at least 1 day apart. The diagnostic criteria were consistent with the Bangladesh National Protocol for the Management of Diabetes and Hypertension.

Pregnant women and patients with life-threatening conditions were excluded from the study. Patients with very high blood pressure or those with acute symptoms of cardiovascular events were referred immediately to tertiary hospitals as per national protocol and were excluded from the analyses. Treatment decisions and patient management followed the Bangladesh national protocol for integrated management of hypertension, diabetes and cholesterol using a total cardiovascular risk approach (online supplemental figure A).

## Digital platform for care coordination

A digital care coordination platform was developed to address challenges in delivering guideline-based decentralised care in primary healthcare settings. A detailed description of the development and deployment of the digital system is available in online supplemental materials (*The development and deployment of a digital platform for primary care hypertension and diabetes care coordination in Bangladesh, online supplemental figures B–D*). Briefly, the digital platform was developed in close collaboration with primary healthcare providers and policymakers through multiple stakeholder meetings and with extensive user-centric research to understand user needs. The system comprised an Android tablet-based application with dashboards to support task shifting/sharing of registering, screening, counselling and routine patient management. The application allowed community health workers and community healthcare providers to enter patient data, which was accessible to healthcare providers at subdistrict-level NCD clinics. With built-in algorithms, the application flags behavioural risk factors, prompts counselling requirements, highlights elevated levels of blood pressure/glucose and generates referral recommendations for confirmatory diagnosis and treatment. At the study NCD clinic, a paramedic performed additional examinations and updated patients' data. The digital care coordination platform generated CVD risk scores based on WHO CVD risk non-laboratory-based charts (South Asia) and had all pertinent information needed by the medical doctor at the NCD clinic to make treatment decisions and management plans, which were then incorporated into the patient record. Patient dashboards were continuously updated, enabling healthcare providers at community clinics to follow-up with routine examinations, medication refills and counselling.

## Data collection and measurements

The present study relied on routinely collected data through the digital care coordination platform. Data were collected during screening and follow-up visits between 1 September 2021 and 28 February 2022. Self-reported sociodemographic characteristics (age, sex, education and religion), health behaviours (tobacco use, physical activity and diet) and medical history (diabetes, CVD, chronic kidney disease, respiratory diseases and cancer) were collected by interviewer-administered questionnaire installed on tablets. Height was measured with stadiometers (Infant Child Adult ShorrBoard, Weight and Measure LLC), and weight was measured with a digital weighing machine (Tanita UM70). Blood pressure was measured with an automatic digital monitor (Omron HEM 7120) following standard operating procedures. Capillary blood was used to measure fasting/random blood glucose with glucometers (Accu-Chek Active).

The primary outcome measurements were changes in the proportion of people with uncontrolled conditions from baseline to follow-up. Patients were considered to have uncontrolled hypertension if they had SBP ≥140 mm Hg and DBP ≥90 mm Hg and uncontrolled diabetes if they had FPG ≥7.0 mmol/L or RPG ≥11.1 mmol/L. Mean changes in blood pressure and blood glucose from baseline were considered secondary outcomes.

Patient retention was assessed as the incidence of dropouts for patients enrolled. Dropout was defined as no recorded visit for ≥2 months. Intervention-related adverse events were documented as safety measurements. This study was approved by the Institutional Review Board of BRAC University (IRB reference number

2020–009-IR). Written informed consent was obtained from all participants.

## Data safety and monitoring

A purpose-built database was developed to facilitate the collection, secure storage and appropriate sharing of study data, including the reporting of clinically relevant results to individual participants. The database was commissioned under the supervision of the Chief Investigator and the respective site Principal Investigator. Personal and clinical data were separated by pseudonymisation to enhance data security. Appropriate authentication and access control mechanisms were implemented. Data were stored securely both locally and in a cloud-based infrastructure. All personal identifiers were removed before analysis.

## Statistical analysis

Descriptive statistics stratified by patient type (hypertension only, diabetes only or having both conditions) were reported. Changes in the proportion of patients with an uncontrolled condition were analysed with generalised estimating equation (GEE) Poisson regression with robust variance. Follow-up visits were included as binary indicators in the regression model to capture the change in blood pressure and blood glucose control relative to the baseline. GEE Poisson is widely used for repeated binary outcomes as it provides consistent estimates of the relative risk and is more stable than the GEE binomial model.[36 37] Change in uncontrolled blood pressure was examined among those with a diagnosis of hypertension (with or without diabetes), while change in uncontrolled blood glucose was evaluated among those with a diagnosis of diabetes, alone or with hypertension. Models were adjusted for age (40–49, 50–59, 60–69, 70–100), sex (male and female), education (no formal education, primary or lower and secondary or higher), religion (Islam and Hindu), BMI category (<18.5, 18.5–22.9 and ≥23 kg/m$^2$) and community clinic location (Khorakhai, Manmathapur and Jahanabad). The analyses were repeated for the subset of patients who had uncontrolled hypertension or diabetes at enrolment.

As secondary outcomes, changes in SBP and DBP in each of the five visits from baseline were assessed with linear mixed-effects models among those with a diagnosis of hypertension (with or without diabetes). Changes in mean fasting and random blood glucose were not assessed given the limited number of repeated measurements (patients had either FPG or RPG, not both). We assessed factors associated with loss to follow-up with Poisson models. A logarithmic time to drop out was incorporated into the incidence ratio estimation. Follow-up time was rounded to a roughly 1-month interval to aid better interpretation. Data management and analyses were conducted using STATA MP v14. Statistical significance was determined by two-sided hypothesis tests (p<0.05).

Sensitivity analyses were conducted to assess the potential bias introduced by sample attrition with inverse probability weighting. The inverse probability weighting approach weighted the analysis by the inverse of the predicted probability of dropping out of the study. A conditional logit model with the same set of covariates described above was used to predict the probabilities. In addition, we restricted our analyses to patients with at least three follow-up visits to test whether the results were sensitive to better balancing sample composition for timing of enrolment and follow-up.

## Patient and public involvement

Before the study, we conducted formative research, including a qualitative assessment to explore technology readiness and acceptability of services among health providers and a quantitative assessment of the readiness of health facilities to deliver NCD services. Multiple meetings with primary care physicians, NCD nurses, community health workers and local health authorities were organised to understand the current care pathway and to identify gaps in usual care. Informal discussions with patients regarding their experience with the current model of care were conducted. Patients were not involved in the analysis or interpretation of the data, nor did they contribute to the writing of the manuscript.

## RESULTS

### Patient characteristics

Of the 2232 people approached, 2208 (98.9%) consented to participate in the study. Among those who consented, 2008 people attended the screening of hypertension and diabetes at the community clinics, of whom 829 were found to have possible hypertension and/or diabetes and were referred to the study NCD clinic for further evaluation (online supplemental figure E). From the 829 referred, 659 attended the study NCD clinic, and among them, 624 were confirmed and included in the analytical sample (mean (SD) age, 59.5 (12.0); 65.1% were female and 45.0% had no formal education). There were 404 patients who had hypertension without diabetes, 75 patients who had diabetes without hypertension and 145 patients who had both. At baseline, of 549 patients with hypertension, 331 (60.3%) had uncontrolled blood pressure, while 163 out of 220 (74.1%) patients with diabetes had uncontrolled glucose levels (table 1).

### Changes in blood pressure and blood glucose

Among the 549 patients with hypertension, the age-adjusted and sex-adjusted proportion of people with uncontrolled blood pressure decreased from 60% (95% CI 56% to 64%) at baseline to 26% (95% CI 19% to 33%) at the third month of follow-up (figure 2), a 56% (incidence risk ratio (IRR) 0.44; 95% CI 0.33 to 0.57) reduction in the probability of having uncontrolled blood pressure after adjustment for covariates (table 2). The proportion of people with uncontrolled blood pressure rebounded to 39% (95% CI 12% to 65%) at the fifth month of follow-up, and the difference was no longer statistically

**Table 1** Patient characteristics at enrolment

| Characteristics* | Entire sample (n=624) | Hypertension only (n=404) | Diabetes only (n=75) | Hypertension and diabetes (n=145) |
|---|---|---|---|---|
| Age in years, mean (SD) | 59.5 (12.0) | 61.5 (12.3) | 55.1 (10.7) | 56.4 (10.6) |
| Age | | | | |
| 40–49 | 136 (21.8) | 68 (16.8) | 28 (37.3) | 40 (27.6) |
| 50–59 | 179 (28.7) | 110 (27.2) | 20 (26.7) | 49 (33.8) |
| 60–69 | 169 (27.1) | 116 (28.7) | 18 (24.0) | 35 (24.1) |
| 70–100 | 140 (22.4) | 110 (27.2) | 9 (12.0) | 21 (14.5) |
| Sex | | | | |
| Female | 406 (65.1) | 274 (67.8) | 41 (54.7) | 91 (62.4) |
| Male | 218 (34.9) | 130 (32.2) | 34 (45.3) | 54 (37.6) |
| Religion | | | | |
| Hindu | 78 (12.5) | 57 (14.1) | 4 (5.3) | 17 (11.7) |
| Islam | 546 (87.5) | 347 (85.9) | 71 (94.7) | 128 (88.3) |
| Education | | | | |
| No formal school | 281 (45.0) | 199 (49.3) | 24 (32.0) | 58 (40.0) |
| Primary or lower | 172 (27.6) | 118 (29.2) | 22 (29.3) | 32 (22.1) |
| Secondary or higher | 171 (27.4) | 87 (21.5) | 29 (38.7) | 55 (37.9) |
| Community clinic | | | | |
| Khorakhai | 204 (32.7) | 120 (29.7) | 33 (44.0) | 51 (35.2) |
| Jahanabad | 202 (32.4) | 132 (32.7) | 23 (30.7) | 47 (32.4) |
| Manmathapur | 218 (34.9) | 152 (37.6) | 19 (25.3) | 47 (32.4) |
| Tabaco use | | | | |
| No | 322 (51.6) | 203 (50.3) | 38 (50.7) | 81 (55.9) |
| Yes | 302 (48.4) | 201 (49.7) | 37 (49.3) | 64 (44.1) |
| Body mass index categories | | | | |
| <18.5 kg/m$^2$ | 68 (10.9) | 55 (13.6) | 6 (8.0) | 7 (4.8) |
| 18.5–22.9 kg/m$^2$ | 243 (38.9) | 159 (39.4) | 26 (34.7) | 58 (40.0) |
| ≥23.0 kg/m$^2$ | 313 (50.2) | 190 (47.0) | 43 (57.3) | 80 (55.2) |
| Hypertension | | | | |
| Baseline BP controlled | 281 (39.7) | 145 (35.9) | n/a | 73 (50.3) |
| Baseline BP uncontrolled† | 331 (60.3) | 259 (64.1) | n/a | 72 (49.7) |
| Diabetes | | | | |
| Baseline glucose controlled | 57 (25.9) | n/a | 15 (20.0) | 42 (29.0) |
| Baseline glucose uncontrolled† | 163 (74.1) | n/a | 60 (80.0) | 103 (71.0) |

*n (%) are presented, unless indicated otherwise.
†Uncontrolled blood pressure is defined as systolic blood pressure ≥140 mm Hg or diastolic blood pressure ≥90 mm Hg, while uncontrolled blood glucose is defined as fasting blood glucose ≥7 mmol/L or random blood glucose ≥11.1 mmol/L.
BP, blood pressure; n/a, not applicable.

significant compared with the proportion at baseline (IRR 0.64; 95% CI 0.32 to 1.25), although the number of observations at the fifth visit (n=20) was notably smaller relative to baseline (n=549). Among the subset of people with uncontrolled blood pressure at baseline, 65% (IRR 0.35; 95% CI 0.26 to 0.47) and 63% (IRR 0.37; 95% CI 0.18 to 0.76) achieved blood pressure control by the third and fifth visits, respectively.

The mean age-adjusted and sex-adjusted SBP at baseline was 145.0 mm Hg (95% CI 143.4 to 146.7) among the 549 people with hypertension. Those with uncontrolled blood pressure had higher mean SBP at baseline (158.7 mm Hg; 95% CI 157.0 to 160.4) compared with the entire sample of patients with hypertension (online supplemental figure F). A substantial reduction in SBP was observed by the first follow-up visit (−8.0 mm Hg; 95%

A  Uncontrolled blood pressure

B  Uncontrolled blood glucose

**Figure 2**  Digital platform designed to support multiple healthcare personnel roles along the care pathway.

CI −10.2 to −5.9) and the reduction sustained throughout the study period (−8.5 mm Hg at 5 months; 95% CI −16.0 to −0.8; online supplemental table A). The reduction in SBP was more pronounced among people with uncontrolled hypertension at baseline (first follow-up visit (−15.2 mm Hg; 95% CI −17.8 to −2.5); at the last follow-up visit (−18.3 mm Hg; 95% CI −27.5 to −9.1)). Similar patterns of estimates were observed for DBP measures.

We observed a similar pattern of change in the control of blood glucose among diabetics. At baseline, 74% (95% CI 68% to 80%) of patients with diabetes had uncontrolled blood glucose. This proportion dropped to 43% (95% CI 32% to 54%) at the third follow-up visit and 31% (95% CI 7.5% to 54.8%) at the fifth follow-up visit. The reduction in the proportion of patients with uncontrolled blood glucose was more pronounced among people with uncontrolled conditions at baseline.

### Patient retention, process evaluation and sensitivity checks
The proportions of patients lost to follow-up immediately after baseline and during the entire study period were 29.1% and 36.2%, respectively (figure 3). Factors associated with an increased probability of loss to follow-up

included having a controlled condition, being a new patient and only having hypertension at baseline (online supplemental table B). Given the nature of continuous enrolment, patients recruited earlier had more follow-up data compared with those who entered the study later.

In total, 41% of follow-up visits were carried out at the community clinics. Average daily numbers of patient visits ranged from 5 to 18 for the NCD clinic and 1–10 in each community clinic (online supplemental figure G). Within the 6-month study period, we recorded no deaths or serious intervention-related adverse events. The results appeared robust in sensitivity analyses incorporating inverse probability weights for dropout and excluding respondents with two or fewer follow-up visits (online supplemental table C).

### DISCUSSION
In this study, we tested a digital technology-supported decentralisation of diabetes and hypertension management in rural Bangladesh. The service delivery model with shifting routine hypertension and diabetes services

| Table 2 | Change in proportion of uncontrolled conditions over follow-up visits | | | |
|---|---|---|---|---|
| | **Blood pressure** | | **Blood glucose** | |
| **Visit** | **Entire sample** | **Baseline uncontrolled cases*** | **Entire sample** | **Baseline uncontrolled cases†** |
| 1 month | 0.63 (0.55, 0.72) | 0.53 (0.47, 0.61) | 0.69 (0.58, 0.82) | 0.57 (0.48, 0.67) |
| 2 months | 0.48 (0.39, 0.60) | 0.38 (0.30, 0.47) | 0.66 (0.54, 0.81) | 0.51 (0.41, 0.63) |
| 3 months | 0.44 (0.33, 0.57) | 0.35 (0.26, 0.47) | 0.60 (0.45, 0.78) | 0.49 (0.38, 0.64) |
| 4 months | 0.49 (0.35, 0.69) | 0.38 (0.27, 0.53) | 0.54 (0.36, 0.80) | 0.41 (0.26, 0.62) |
| 5 months | 0.64 (0.32, 1.25) | 0.37 (0.18, 0.76) | 0.44 (0.20, 0.96) | 0.16 (0.05, 0.54) |
| Constant | 0.44 (0.30, 0.64) | 0.99 (0.93, 1.05) | 1.04 (0.67, 1.60) | 1.33 (0.98, 1.78) |
| N | 549 | 331 | 220 | 163 |

*Baseline uncontrolled cases were defined patients with systolic blood pressure ≥140 mm Hg or diastolic blood pressure ≥90 mm Hg.
†Baseline uncontrolled cases were defined patients with fasting plasma glucose ≥7.0 mmol/L or random plasma glucose ≥11.1 mmol/L.
‡Models adjusted for age (40–49, 50–59, 60–69 and 70–100), sex (male and female), education (no formal education, primary or lower and secondary or higher), religion (Islam and Hindu), body mass index category (under, normal, overweight or obese), community clinics (Khorakhai, Manmathapur, Jahanabad).

A Entire sample

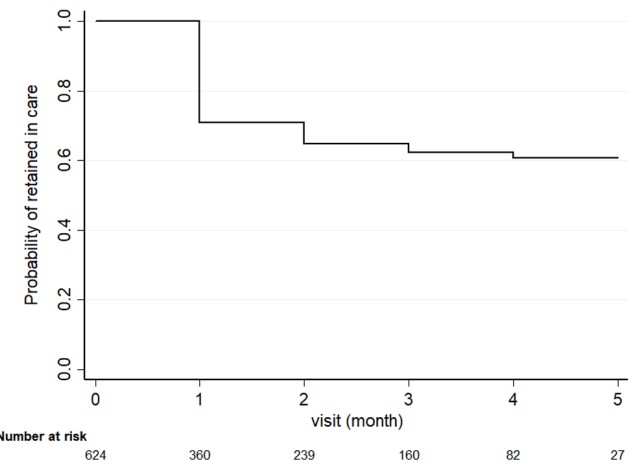

B By baseline conditions

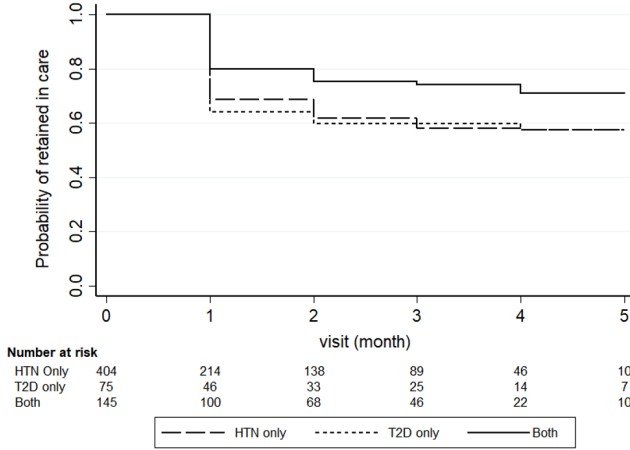

**Figure 3** Change in the proportion of uncontrolled conditions over follow-up visits. Results adjusted age and sex. Uncontrolled blood pressure is defined as systolic blood pressure ≥140 mm Hg or diastolic blood pressure ≥90 mm Hg, while uncontrolled blood glucose is defined as fasting blood glucose ≥7 mmol/L or random blood glucose ≥11.1 mmol/L. HTN, hypertension; T2D, type 2 diabetes.

to lowest-level primary care facilities and non-physician health workers achieved a high treatment success rate and relatively high patient retention. The results highlighted the potential of technology-supported decentralised care to address the critical shortage of health service providers and geographical barriers to access, along with the retention of patients with hypertension and diabetes.

Primary care delivery of hypertension and diabetes services in Bangladesh is hampered by a critical shortage of trained health workers and poor geographical accessibility. In Bangladesh, only about one in three patients with hypertension or diabetes is currently on treatment.[28 29] The government NCD Corners at the subdistrict level are remote and poorly accessible to a large body of population living with NCDs.[13] As a result, the majority of people with hypertension and diabetes are dependent on the largely unregulated private sector and frequently receive services from untrained or unqualified care providers.[38]

Our preliminary results showed that with the support of a digital care coordination platform and proper training, non-physician health workers based in the community or community clinics can successfully deliver screening for hypertension and diabetes, routine follow-up, counselling, drug refills and other aspects of NCD management. This approach to healthcare delivery has the potential to substantially improve access to healthcare and reduce the burden on physician-managed NCD clinics. Overall, the proportion of patient follow-up visits carried out at community clinics was 41%. Routine patient follow-up at community locations could be further enhanced by introducing telemedicine and dose titration by non-physician health workers. Through technology-supported decentralisation and the involvement of a large number of peripheral primary care facilities, hypertension and diabetes services could be rapidly expanded in Bangladesh and similar global settings.

In addition to the lack of human resources, inconsistent adherence to treatment protocols among healthcare providers, poor patient management and irregular follow-up are important obstacles to effective hypertension and diabetes management.[13 39] As a result of these barriers, compounded by low levels of awareness and screening, the proportion of all patients with the controlled condition is merely 7% for diabetes and 12% for hypertension in Bangladesh.[29 30] A variety of digital tools have been examined for hypertension and diabetes care quality improvement in South Asian settings with mixed findings.[23 24 40 41] Digital tools can be invaluable in strengthening health systems for NCD care delivery; however, they need to be an integral part of a multifaceted intervention, with thoughtful consideration of care pathways and user interfaces, to exert effects within specific health system contexts.[6] Following international standards of interoperability throughout the designing phase, the digital platform resources developed in the present study are open-source and can be readily used by the global health community.

Treatment adherence and retention are major challenges in NCD management, particularly in LMICs.[42–44] In a recent study of task shifting to non-physician clinicians for integrated management of hypertension and diabetes in rural Cameron, Labhardt and colleagues reported a retention rate of 18.1% among treated patients at 1 year. Another community health worker-led, technology-enabled private sector intervention for diabetes and hypertension management among the urban poor in India observed a 44.0% retention rate at 6 months.[44] The observed retention rates of 63.8% by the end of the present study appeared high. While the reasons were not formally tested, possible explanations include the ease of obtaining medication refills, alleviated travel-related burden and more counselling time with non-physician health workers.

Our findings showed several factors were associated with loss to follow-up, including being newly diagnosed,

having a controlled condition at enrolment and/or being diagnosed with hypertension only. In addition, the results showed a particularly high attrition rate immediately after enrolment, consistent with previous studies.[42 44] Attrition incidence appeared to decrease with longer follow-up despite higher accumulative attrition rates over time, likely due to the depletion of individuals with a high propensity of dropping out. These observations may have important implications for routine healthcare delivery to improve patient retention. Additional efforts may be needed to emphasise the need for regular follow-up with newly diagnosed patients, those with controlled status at enrolment and those detected with hypertension only, especially at enrolment.

As a proof-of-concept study, the project comprised an uncontrolled and unblinded evaluation of a decentralised service model, supported by a digital care coordination platform. Nevertheless, the observed rates of 61% for blood pressure control and 69% for blood glucose control at the end of the study compared favourably with existing implementation studies and clinical trials of multicomponent health system interventions in developing countries.[45–48] For example, in a recent cluster-randomised controlled trial in Bangladesh, India and Sri Lanka, a multifaceted intervention involving trained community health workers for home-based blood pressure monitoring and counselling achieved blood pressure control in 53.2% of the participants in the intervention group and 43.7% of the control group.[46] The findings will inform the design of our planned further studies to test the feasibility and utility of the interventions within the government primary care system, as well as to evaluate the effectiveness of the digital technology-supported decentralised care in Bangladesh.

## CONCLUSION

In the proof-of-concept study, a digital technology-supported decentralised care model achieved high levels of blood pressure and blood glucose control among patients with hypertension and diabetes in Bangladesh. Our results suggest that with proper training, supportive supervision and access to digital technology, non-physician health workers at peripheral primary facilities can deliver health screening, counselling, routine drug refills and other aspects of patient management for people with hypertension and diabetes. The present study highlighted the potential to involve peripheral primary care facilities and community health workers to rapidly expand much-needed hypertension and diabetes services in Bangladesh and similar global settings. Further investigations are needed to evaluate the effectiveness of decentralised hypertension and diabetes care.

**Author affiliations**
[1]Lee Kong Chian School of Medicine, Nanyang Technological University, Singapore
[2]Centre for Non-communicable Diseases and Nutrition, BRAC University James P Grant School of Public Health, Dhaka, Bangladesh
[3]Tyree Foundation Institute of Health Engineering, UNSW, Sydney, New South Wales, Australia
[4]Baker Heart and Diabetes Institute, Melbourne, Victoria, Australia
[5]Sprightly Pte Ltd, Singapore
[6]Google Health, Palo Alto, California, USA
[7]School of Psychology and Public Health, La Trobe University, Melbourne, Victoria, Australia
[8]Department of Epidemiology and Biostatistics, Imperial College London, London, UK

**Acknowledgements** The authors would like to thank Dr. Theresia Handayani Mina and Dr. Benjamin Lam Chih Chiang for their thoughtful comments on the manuscript.

**Contributors** MKM, JC, IG, BO, RRP and FH contributed to the conception and design of the study. IG, AR, FH, MKM, RRP, MMH and AA worked on the development of the digital platform. RRP, AA, MKM and MMH contributed to the implementation of data collection and management. WX implemented the analysis of the data and wrote the first draft of the manuscript. All authors contributed to the interpretation of the data and the revision of the manuscript.

**Funding** This research was funded by the National Institute for Health Research (NIHR) (16/136/68). The funder had no role in study design, data collection and analysis, decision to publish, or preparation of the manuscript.

**Competing interests** None declared.

**Patient and public involvement** Patients and the public were involved in the design and conduct, but not in reporting or dissemination plans of this research. Refer to the Methods section for further details.

**Patient consent for publication** Consent obtained directly from patient(s).

**Ethics approval** This study involves human participants and was approved by Institutional Review Board of BRAC University (IRB reference number 2020-009-IR). Participants gave informed consent to participate in the study before taking part.

**Provenance and peer review** Not commissioned; externally peer reviewed.

**Data availability statement** Data are available upon reasonable request. Original data are available upon reasonable request. Please contact the corresponding author for further information (malay.mridha@bracu.ac.bd).

**ORCID iDs**
Md Mokbul Hossain http://orcid.org/0000-0002-8058-4776
Malay Kanti Mridha http://orcid.org/0000-0001-9226-457X

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
