## [Reviewer comments · BMJ Open]

This paper was submitted to a another journal from BMJ but declined for publication following peer review. The authors addressed the reviewers' comments and submitted the revised paper to BMJ Open. The paper was subsequently accepted for publication at BMJ Open.

ARTICLE DETAILS

TITLE (PROVISIONAL)	Enhancing Care Quality and Accessibility through Digital Technology-Supported Decentralization of Hypertension and Diabetes Management: A Proof-of-Concept Study in Rural Bangladesh
AUTHORS	Xie, Wubin; Paul, Rina Rani; Goon, Ian; Anan, Aysha; Rahim, Aminur; Hossain, Md Mokbul; Hersch, Fred; Oldenburg, B; Chambers, John; Mridha, Malay Kanti

VERSION 1 – REVIEW

REVIEWER	Walley, John University of Leeds, Nuffield Centre for International Health and Development, Leeds Institute of Health Sciences
REVIEW RETURNED	12-May-2023

GENERAL COMMENTS	This is an impressive study and article. I recommend acceptance. I think it adds substantively to the experience of delivering NCD care in LMICs. I'm not suggesting requiring revisions, but if you are asked to do so, then consider: Methods: Rephrase/add for clarity 'Basic physical examination by the community health clinic providers.. to make clear if this included a glucometer/ glucose test by the community health clinic providers. Typo: p18. L 10-13 Controll => control.
---

REVIEWER	Morelli, Daniela Moraes Instituto de Efectividad Clínica y Sanitaria
REVIEW RETURNED	25-May-2023

GENERAL COMMENTS	The manuscript is very well organized and successfully synthesizes information about a complex intervention in a way that makes reading enjoyable. This study offers evidence that may contribute to improving access to care in rural Bangladesh, as well as a way to assess the impact of this decentralized approach on the quality and control of hypertension and diabetes, and may be especially useful for understanding that alternative processes,
--

	such as decentralization, can improve access to care in remote regions. Minor comments Abstract Page 5 – Row 18: It is requested to include the study design. Strength and limitation of the study Page 6 – Row 5: The authors highlight this study as the first of its kind. There are plenty of similar studies evaluating a decentralised intervention using digital health in the area of chronic diseases. Please clarify if this is one of the pioneer studies in rural areas of the region or the country. Introduction Page 9 – Row 5: Describing it as a proof of concept is not clear for the average reader. It would be important to briefly clarify what this definition represents, to facilitate understanding by readers. Kendig CE. What is Proof of Concept Research and how does it Generate Epistemic and Ethical Categories for Future Scientific Practice? Sci Eng Ethics. 2016 Jun;22(3):735-53. doi: 10.1007/s11948-015-9654-0. Epub 2015 May 26. PMID: 26009258 Methods Study Design Page 11 – Row 6: The design of the study is a description of a single cohort and should be clarified in the methods section. Statistical analysis Page 16 – Row 29: The authors mention that given the nature of continuous enrolment, patients recruited earlier had a lengthier follow-up compared with those entered the study later. This needs to be discussed further and clarify if the GEE Poisson model used included the exposure time of each individual in order to correctly estimate the IRR. This does not generate a bias if the follow-up time of each one was included in the Poisson analysis. Results Page 18 – Row 53: The authors report a 56% reduction in the probability of having uncontrolled blood pressure after adjustment for covariates (from 60% at baseline to 26%) at the third month of follow-up, but this group returns to the baseline proportion of poor control at 6 months. Please add a brief comment about this finding. Page 19 – Row 6: The authors mention that the attrition rates immediately after baseline and during the entire study period were 29.1% and 36.2% respectively, which are quite high for a short term cohort. Please add a brief comment about the potential bias induced by this proportion of patients lost to follow-up.
--	--

VERSION 1 – AUTHOR RESPONSE

Comments from the Reviewer #1

Comment 1.1 This is an impressive study and article. I recommend acceptance. I think it adds substantively to the experience of delivering NCD care in LMICs. I'm not suggesting requiring revisions, but if you are asked to do so, then consider: Methods: Rephrase/add for clarity 'Basic physical examination by the community health clinic providers...' to make clear if this included a

glucometer/ glucose test by the community health clinic providers. Typo: p18. L 10-13 Controll => control.

Response. We appreciate the reviewer's favorable evaluation of our study. Suggested changes have been made. Particularly, we have clarified the procedure of basic physical examination (copied below) and corrected the typo.

"At the follow-up visits, community healthcare providers perform a basic physical examination (i.e., body weight and height measurement, calculation of BMI, measure blood pressure and blood glucose measurement), inquire about any new symptoms, assessed CVD risk, emphasize the importance of medication adherence, provide advice regarding a healthy lifestyle, and refill prescribed medications."

Comments from the Reviewer #2

Comment 2.1 The manuscript is very well organized and successfully synthesizes information about a complex intervention in a way that makes reading enjoyable. This study offers evidence that may contribute to improving access to care in rural Bangladesh, as well as a way to assess the impact of this decentralized approach on the quality and control of hypertension and diabetes and may be especially useful for understanding that alternative processes, such as decentralization, can improve access to care in remote regions.

Response. We appreciate the reviewer's overall favorable review of our work.

Minor comments

Abstract

Comment 2.2 Page 5 – Row 18: It is requested to include the study design.

Response. Thanks for pointing out the omission. The requested information has been added:

"Design and setting The study is a single cohort proof-of-concept study. The key interventions comprised shifting screening, routine monitoring, and dispensing of medication refill from a doctor-managed subdistrict NCD clinic to non-physician health worker managed village level community clinics..."

Strength and limitation of the study

Comment 2.3 Page 6 – Row 5: The authors highlight this study as the first of its kind. There are plenty of similar studies evaluating a decentralised intervention using digital health in the area of chronic diseases. Please clarify if this is one of the pioneer studies in rural areas of the region or the country.

Response. It is true that many studies have explored the idea of task sharing/shifting in NCD care in LMIC context with or without digital health component, such as those included in the systematic review by Anand et al.¹ A number of studies have investigated interventions such as shifting lifestyle modification counselling to nurses or pharmacist, or engaging community health workers for home-based blood pressure monitoring and life-style reinforcing, and algorithm-based management with or without electronic decision support system. However, these studies have typically evaluated task sharing within the same healthcare facilities^{2,3} rather than coordinated task sharing among different levels of primary care. Other research has explored the integration of community-based care delivery with single disease focus or without digital care coordination component (e.g., the COBRA-BPS study⁴ and the HOPE-4 study).⁵

We are not aware of other studies that have formally tested either 1) the idea of combining community health workers and grass-root level primary care facilities for NCD care delivery (decentralization) with digital care coordination for improved task sharing and referral, or 2) the potential of using decentralized care to expand NCD care coverage. Our study demonstrates how technology can support integrated care between community and facility through a task based coordinated care approach. Nevertheless, we do recognize that none of the individual components of our proposed intervention are entirely new, so we have revised the following sentence accordingly:

“The study is among the first to evaluate a digital technology-supported, decentralized primary care model for integrated diabetes and hypertension management in a resource-constrained setting.”

Introduction

Comment 2.4 Page 9 – Row 5: Describing it as a proof of concept is not clear for the average reader. It would be important to briefly clarify what this definition represents, to facilitate understanding by readers.

Kendig CE. What is Proof of Concept Research and how does it Generate Epistemic and Ethical Categories for Future Scientific Practice? *Sci Eng Ethics*. 2016 Jun;22(3):735-53. doi: 10.1007/s11948-015-9654-0. Epub 2015 May 26. PMID: 26009258

Response. While we agree that the term “proof-of-concept” is perhaps not used as widely as “feasibility” or “pilot”,⁶ we believe that ‘proof-of-concept’ is more appropriate to describe our trial as our study specially aims to investigate a novel intervention. Accordingly, we have added a few details to clarify our study as a proof-of-concept, small-scale investigation:

“In the present proof-of-concept study, we aimed to 1) generate preliminary data on the effectiveness of a novel digital technology supported decentralization of primary care diabetes and hypertension management in a rural subdistrict in Northern Bangladesh, and 2) to understand its potential future implications for the patient care pathway, the patient load in primary care facilities at a subdistrict and village level, and patient retention.”

Methods

Study design

Comment 2.5 Page 11 – Row 6: The design of the study is a description of a single cohort and should be clarified in the methods section.

Response. We have made the recommended change:

“The study is a single cohort proof-of-concept study. Decentralized hypertension and diabetes care comprised screening, routine monitoring, and dispensing of ...”

Statistical analysis

Comment 2.6 Page 16 – Row 29: The authors mention that given the nature of continuous enrolment; patients recruited earlier had a lengthier follow-up compared with those entered the study later. This needs to be discussed further and clarify if the GEE Poisson model used included the exposure time of each individual in order to correctly estimate the IRR. This does not generate a bias if the follow-up time of each one was included in the Poisson analysis.

Response. Exposure time was included as the key independent variable to quantify the change in blood pressure and blood glucose control using binary dummy variables for each follow-up time point relative to the baseline. We added this information in the statistical analysis section.

“Changes in proportion of patients with an uncontrolled condition were analysed with generalized estimating equation (GEE) Poisson regression with robust variance. Follow-up visit was included as binary indicators in the regression model to capture the change in blood pressure and blood glucose control relative to the baseline.”

Results

Comment 2.7 Page 18 – Row 53: The authors report a 56% reduction in the probability of having uncontrolled blood pressure after adjustment for covariates (from 60% at baseline to 26%) at the third month of follow-up, but this group returns to the baseline proportion of poor control at 6 months. Please add a brief comment about this finding.

Response. The proportion of patients with uncontrolled blood pressure at 6 month was 39% as compared with 60% at baseline (Figure 2). The reviewer is right that although there was a 36% difference, it was not statistically significant (IRR 0.64; 95% CI 0.32, 1.25). We have added a sentence to clarify this finding.

“Amongst the 549 patients with hypertension, the age- and sex-adjusted proportion of people with uncontrolled blood pressure decreased from 60% (95% CI, 56%, 64%) at baseline to 26% (95% CI, 19%, 33%) at the third month of follow-up (Figure 2), a 56% (IRR 0.44; 95% CI, 0.33, 0.57) reduction in the probability of having uncontrolled blood pressure after adjustment for covariates (Table 2). The proportion of people with uncontrolled blood pressure rebounded to 39% (95% CI, 12%, 65%) at the fifth month of follow-up, and the difference was no longer statistically significant compared with the proportion at baseline (IRR 0.64; 95% CI, 0.32, 1.25), although the number of observations at the fifth visit (n=20) was notably smaller relative to baseline (n=549).”

Comment 2.8 Page 19 – Row 6: The authors mention that the attrition rates immediately after baseline and during the entire study period were 29.1% and 36.2% respectively, which are quite high for a short-term cohort. Please add a brief comment about the potential bias induced by this proportion of patients lost to follow-up.

Response. It is widely recognized that patient retention is a tremendous challenge in hypertension and diabetes care largely due to the fact that these conditions are often asymptomatic and not immediately life-threatening. The present study was designed to test the modified care pathways and patient retention in a real-world setting rather than a highly controlled context such as with an efficacy trial. As elaborated in the discussion, the observed rates of dropping out of the study compare favorably with previous implementation studies.^{7,8} To evaluate the potential biases introduced by sample attrition, we conducted sensitivity analyses (results shown in Supplementary Table 3), estimating the key results with inverse probability weighting using the predicted probability of dropping out from the study from a conditional logit model with the same set of covariates, as well as restricting the analyses to patients with 3 or more follow-up visits, and the results appeared to be robust.

While the robustness of our results to potential sample attrition bias was examined, we recognize that the discussion section could be strengthened with a brief discussion on the factors associated with attrition and the decreasing attrition incidence. A paragraph has now been added in the discussion section for this purpose.

“Our findings showed several factors were associated with loss to follow-up, including being newly diagnosed, having a controlled condition at enrolment, and/or being diagnosed with hypertension only. In addition, the results showed a particularly high attrition rate immediately after enrolment, consistent with previous studies.^{7,8} Attrition incidence appeared to decrease with longer follow-up despite higher cumulative attrition rates over time, likely due to the depletion of individuals with high propensity of dropping out. These observations may have important implications for routine healthcare delivery to improve patient retention. Additional efforts may be needed to emphasize the need for regular follow-up to newly diagnosed patients, those with controlled status at enrolment, and those detected with hypertension only, especially at enrolment.”